

# Creep of CarbFix Basalt: Influence of Rock-fluid Interaction

Tiange Xing[1], Hamed O. Ghaffari[1], Ulrich Mok[1], Matej Pec[1]

[1]Department of Earth, Atmospheric and Planetary Sciences, Massachusetts Institute of Technology

*Correspondence to*: Tiange Xing (tiange@mit.edu)

**Abstract.** Geological carbon sequestration provides permanent $CO_2$ storage to mitigate the current high concentration of $CO_2$ in the atmosphere. $CO_2$ mineralization in basalts has been proven to be one of the most secure storage options. For successful implementation and future improvements of this technology, the time-dependent deformation behavior of reservoir rocks in presence of reactive fluids needs to be studied in detail. We conducted load stepping creep experiments on basalts from the CarbFix site (Iceland) under several pore fluid conditions (dry, $H_2O$-saturated and $H_2O+CO_2$-saturated) at temperature, T≈80°C and effective pressure, $P_{eff}$ = 50 MPa, during which we collected mechanical, acoustic and pore fluid chemistry data. We observed transient creep at stresses as low as 11% of the ultimate failure strength, well below the stress level at the onset of bulk dilatancy. Acoustic emissions (AEs) correlated strongly with strain accumulation, indicating that the creep deformation was a brittle process in agreement with microstructural observations. The rate and magnitude of AEs were higher in fluid-saturated experiments than in dry conditions. We infer that the predominant mechanism governing creep deformation is time- and stress-dependent sub-critical dilatant cracking. Our results suggest that the presence of aqueous fluids exerts first order control on creep deformation of basaltic rocks, while the composition of the fluids plays only a secondary role under the studied conditions.

## 1 Introduction

The concentration of atmospheric $CO_2$ has seen a significant increase over the last century, raising concerns about the more frequent occurrence of extreme weather, sea-level rise and the projected increase of average global temperature (Broecker, 1975). It is estimated that about 800 Gt $CO_2$ will need to be stored by the end of the century to keep the global temperature increase below 1.5 °C compared to pre-industrial levels (National Academies of Sciences, Engineering, 2019). Such large volumes can practically be stored in the sub-surface. Geological carbon sequestration (GCS) by in-situ carbon mineralization is recognized as one of the most secure, long-term storage solutions (Gislason and Oelkers, 2014; Kelemen and Matter, 2008; Lackner et al., 1995; Mani et al., 2008; Seifritz, 1990; Snæbjörnsdóttir et al., 2020). To date, several pilot projects have been launched to study GCS in basalt reservoirs, including the CarbFix program in Iceland (Callow et al., 2018; Gislason et al., 2010; Oelkers et al., 2008; Snæbjörnsdóttir et al., 2018) and the Wallula basalt (part of Columbia River Basalt Group) sequestration project in Washington, US (McGrail et al., 2006, 2011, 2017; Zakharova et al., 2012).

GCS involves the injection of fluids, either supercritical $CO_2$ or $CO_2$ in an aqueous solution, into the formations. Basalts are composed of mafic minerals such as pyroxene (($Mg,Fe)_2Si_2O_6$), plagioclase (($Ca,Na)Al_{1.70}Si_{2.30}O_8$), and olivine (($Mg,Fe)_2SiO_4$) as well as mafic glass, which react with $CO_2$ to form carbonate minerals (e.g $MgCO_3$, $CaCO_3$,



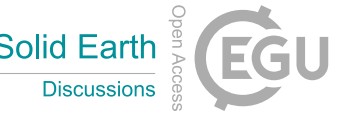

FeCO$_3$ etc.), thus binding the injected CO$_2$ in mineral structure (Gislason and Hans, 1987; Hangx and Spiers, 2009;
Matter et al., 2007; Oelkers et al., 2008). Carbonation reactions appear to be rapid in natural conditions; more than
95% of the CO$_2$ injected into the CarbFix site in Iceland was converted to carbonate minerals in less than 2 years
(Matter et al., 2016). Relevant fluid and mineral reactions can be formulated as follows (Hangx and Spiers, 2009;
Hansen et al., 2005; Kelemen and Matter, 2008; Oelkers et al., 2008):
Dissociation:
$$CO_2 + H_2O \rightleftharpoons H_2CO_3 \tag{Eq. 1}$$

$$H_2CO_3 \rightleftharpoons HCO_3^- + H^+ \tag{Eq. 2}$$

Dissolution:
$$(Mg, Fe)_2Si_2O_6 + 4H^+ = 2(Mg, Fe)^{2+} + 2SiO_2 + 2H_2O \tag{Eq. 3}$$

$$CaAl_2Si_2O_8 + 8H^+ = Ca^{2+} + 2Al^{3+} + 2SiO_2 + 4H_2O \tag{Eq. 4}$$

$$(Mg, Fe)_2SiO_4 + 4H^+ = 2(Mg, Fe)^{2+} + SiO_2 + 2H_2O \tag{Eq. 5}$$

Precipitation:
$$(Mg, Ca, Fe)^{2+} + HCO_3^- = (Mg, Ca, Fe)CO_3 + H^+ \tag{Eq. 6}$$

CO$_2$ is dissolved in water to form an acidic solution (Eq. 1-2). The rocks dissolve to liberate divalent cations (Eq. 3-
5), which upon saturation in the fluid, precipitate as carbonate minerals (Eq. 6) further downstream from the injection
site.
The mechanical and transport behavior of rocks can be significantly affected during GCS by the interaction between
rock and fluid, both from a mechanical as well as chemical perspective ( Baud et al., 2000; Dunning & Miller, 1985;
Heard, 1960; Helmons et al., 2016; Rutter & Hackston, 2017). The mechanical effect of pore fluid is readily accounted
for by using the effective pressure law (Terzaghi, 1943). The pore fluid acts against the normal stresses acting on crack
surfaces hence reducing the shear stress necessary to overcome internal friction of the rocks. Increase in pore pressure
during injection can trigger seismicity and therefore pore pressure has to be carefully monitored (Atkinson et al., 2020;
Guglielmi et al., 2015). In addition to this mechanical effect, a number of chemical processes can occur in the presence
of reactive fluids, leading to complex coupling between processes. For example, the replacement of mafic minerals
with carbonates can result in an up to ~44% increase in solid molar volume (Goff and Lackner, 1998; Hansen et al.,
2005; Kelemen and Matter, 2008) potentially clogging pore space, reducing permeability and increasing pore pressure.
Alternatively, this volume expansion can  generate stresses causing reaction-induced fracturing, which provides
additional fluid pathways and maintains porosity and permeability for the reaction to proceed (Iyer et al., 2008;
Jamtveit et al., 2009; Kelemen & Matter, 2008; Lambart et al., 2018; Macdonald & Fyfe, 1985; Renard et al., 2020;
Rudge et al., 2010; Skarbek et al., 2018; Xing et al., 2018; Zhu et al., 2016). The fracturing behavior itself is affected
by the fluid chemistry via kinetic reduction of fracture energy due to fluid absorption on mineral surfaces and crack
tip blunting (Baud et al., 2000; Orowan, 1944; Rutter, 1972; Scholz, 1968), and activation of fluid-promoted stress
corrosion processes such as subcritical crack growth resulting in time-dependent deformation, which is the focus of



this paper (Anderson & Grew, 1977; Atkinson, 1984; Atkinson & Meredith, 1987; Brantut et al., 2013; Nara et al.,
2013; Rice, 1978).
This time-dependent deformation, often called "brittle creep" or "static fatigue", has been observed in all types of
rocks tested to date (Atkinson & Meredith, 1987; Brantut et al., 2012; Kranz et al., 1982; Robertson, 1960; Scholz,
1968; Zhang et al., 2012). During brittle creep, flaws such as micro-cracks contained in natural rocks are sub-critically
stressed and propagate slowly due to stress corrosion (a chemical weakening process) at crack tips in the presence of
fluids. Sample-scale fracture then occurs after some time delay when the cracks coalesce and reach a critical length.
As a result, the rocks lose their load bearing capabilities and fail along a macroscopic fault plane at stresses well below
their short term strength (Scholz, 1972). For the sake of simplicity, we will use creep in the following text to refer to
this brittle creep deformation.
It has been shown by experiments, observations and modelling that stress corrosion is the dominant mechanism of
subcritical crack growth in rocks under upper crustal conditions (Brantut et al., 2012; Michalske and Freiman, 1983;
Reber and Pec, 2018). Brittle creep deformation can be accelerated due to changes in the rate of stress corrosion
induced by the chemistry of the injected fluids (Renard et al., 2005, 2020) or decelerated by crack tip blunting due to
fluid interaction (Scholz, 1968). Overall, it is hypothesized that changes in stress corrosion crack growth rate due to a
change in fluid chemistry will be reflected in similar changes of the macroscopic creep strain rate, either accelerating
or decelerating based on the details of the ongoing dissolution – precipitation reactions (Brantut et al., 2013). Hence,
the effect of $CO_2$-rich fluids needs to be quantified for GCS applications.
To summarize, the influence of rock-fluid interaction on deformation is complicated and includes the coupled effects
of mineral dissolution and precipitation, kinetics of fluid assisted deformation and injection pressure built-up, finally
resulting in time-dependent rock deformation. Carbonation changes the bulk composition of the basalts, alters their
strength and pore structure, and affects the permeability of the rocks (Dunkel et al., 2017; Kanakiya et al., 2017;
Kelemen et al., 2013; Kelemen & Hirth, 2012; Lisabeth et al., 2017; Xing et al., 2018; Zhu et al., 2016). Understanding
of the effects of rock-fluid interaction on deformation requires dedicated laboratory studies with diverse fluid
compositions at in-situ pressure conditions and at elevated temperatures acting over extended timescales. The present
study aims at elucidating the effect of rock-fluid interaction on the time-dependent rock deformation by investigating
long-term creep of Iceland Basalt saturated with various fluid compositions.
**2 Materials and Methods**
**2.1 Starting Material and Sample Configuration**
We used Iceland Basalt drill cores from the CarbFix site, collected at ~350 m depth. The composition of Iceland Basalt
has been identified as tholeiite and contains ~ 25 wt% of calcium, magnesium and iron oxides (7-10 wt% Ca; 5-6 wt%
Mg; 7-13 wt% Fe) with an average porosity of ~ 8% based on hydrological and tracer recovery modeling (Alfredsson
et al., 2008, 2013; Aradóttir et al., 2012; Matter and Kelemen, 2009; Snæbjörnsdóttir and Gislason, 2016). The rock
is formed by an aphanitic matrix that consists of crystals of feldspars, clinopyroxene, iron ore and glass. The fraction
of crystal-to-glass ratio as well as crystal habitat is variable as documented in Figure 1. Round pores with a mean





diameter of ~ 0.5 mm are randomly distributed throughout the matrix, some are filled with feldspar (primarily
potassium feldspar) and some are voids with no filling (Figure 1). Pore-, as well as pre-existing crack-walls are coated
by a thin layer of a phyllosilicate as documented in Figure 1d and 1e. The matrix is locally altered by dissolution of
larger subhedral feldspar crystals and local replacement by phyllosilicate (see Figure 1b and 1e). Cylindrical samples
were ground to ~ 40 mm in diameter and ~ 80 mm in length (see Table 1). The samples were jacketed using copper
foil of ~0.05 mm thickness, joined to titanium end-caps by Viton tubes and coated with Duralco 4538 epoxy. The end-
caps had a concentric hole which allows fluid access to the sample. Figure 2 shows the schematics of the sample
configuration in this study. An internal force gauge was mounted below the sample inside the vessel, allowing direct
measurement of the differential stress ($\Delta\sigma = \sigma_1 - \sigma_3$). Displacement of the axial piston was measured externally using
a linear variable differential transformer (LVDT). Variations of the sample length were measured using two internal
LVDTs. Local axial ($\epsilon_a$) and radial strains ($\epsilon_r$) of the rock were measured using strain gauges affixed to the copper
jacket around the sample. Piezoelectric sensors were installed around the sample for passive monitoring of acoustic
emissions (AE).

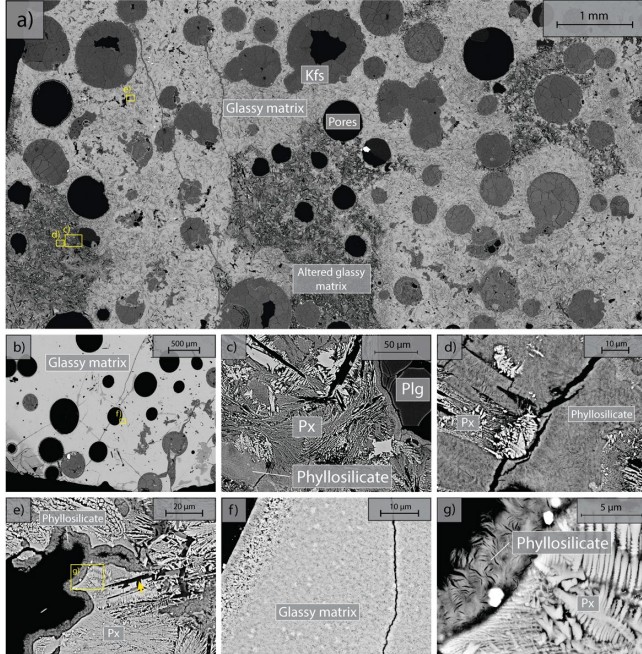


**Figure 1**. Backscattered electron (BSE) images of the starting material. Location of higher-magnification images is
shown by yellow rectangles a) Representative sample microstructure; Glassy matrix shows a range of crystal content
and habitat and is locally altered; b) Another common glassy matrix microstructure with finer, more homogenous
crystals; c) Detail of altered matrix and pore wall. Note black voids in the shape of subhedral feldspar laths in the
matrix; d) Detail of patchy phyllosilicate alteration and dendritic crystals in altered matrix; e) high-magnification
image of dendritic crystals forming the matrix and phyllosilicate coating of the pore wall; f) high magnification image
of glassy matrix with homogenous small crystals g) high-magnification image of dendritic crystals forming the
unaltered matrix and  high magnification image of the phyllosilicate alteration.



To minimize the issue of inter-sample variability, we adopted the 'stress-stepping' experimental procedures to study
creep deformation (Heap et al., 2009; Lockner, 1993). Piezoelectric sensors allowing independent recording of
compressional and shear waves were fabricated with PZT-5A ceramics with thickness of ~3 to 5 mm and resonance
frequency of ~450 kHz to 1 MHz. The PZT-5A crystals were mounted on titanium spacers with one side concavely
curved to match the sample surface, thus providing protection of the sensing crystals and optimal contact area. A back-
up element was epoxied to the back of the sensor to minimize ringing. We also used analog low pass filters (~500
kHz) compatible with the frequency range of the employed PZT ceramics to reduce the electromagnetic interference
(EMI) effect. Data was collected using two combined 4-channel universal serial bus (USB) oscilloscopes, recording
at 50 MS/s with a 12-bit resolution (TiePie HS4-50). Using low noise amplifiers (ITASCA-60dB), we carefully
selected the most sensitive sensor positions, preferably far from each other, as master channels. The data collection
system was set such that, if the master channels detected a signal satisfying a sufficiently large signal/noise ratio in a
moving time window, the event would be recorded in all channels. We amplified the two master channels with a flat
gain of 60 dB in a frequency range of 50 kHz to 1.5 MHz. Frequencies from 1.5 MHz to 15 MHz were amplified
nonlinearly, the gain decreasing exponentially from 52 dB to 37 dB with increasing frequency (Ghaffari and Pec,
2020). Considering the above limitations, the main frequency range of the recording system was between ~50 and
500kHz, although other frequencies could be recorded owing to the exponential nature of the amplification filters.

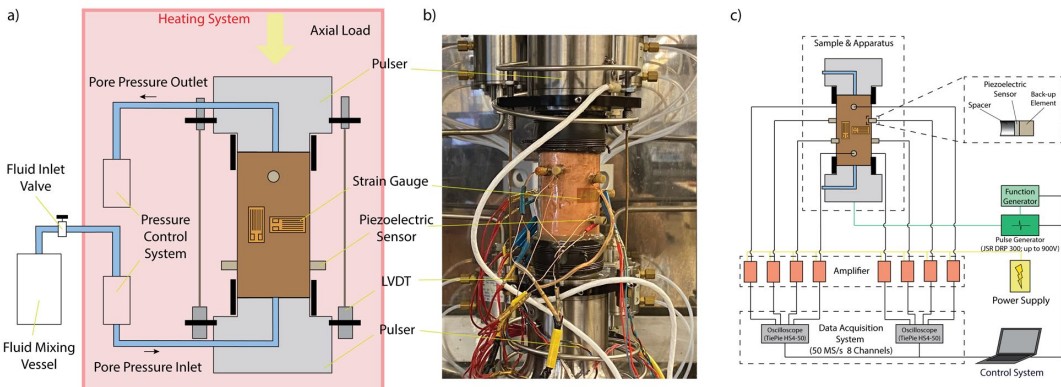


**Figure 2**. a) Schematics of sample configuration. The whole sample assembly and pore fluid actuators are enclosed
in a servo-controlled heating system to ensure a uniform temperature condition; b) Photo of the sample assembly; c)
Illustration of the acoustic emission recording system.
**2.2 Experimental Setup and Analytical Methods**
All experiments were conducted at 50 MPa effective pressure, $P_{eff}$, with pore fluid pressures, $P_f$, of either 0 or 5 MPa
for dry and fluid-saturated experiments, respectively. The fluids used in this study were $H_2O$ and $H_2O+CO_2$. The fluid-
saturated samples were first immersed in deionized water under vacuum for more than 30 days prior to the experiment.
Details of the experimental conditions are listed in Table 1. The samples were inserted in the NER Autolab 3000
testing rig installed at MIT and deformed under tri-axial stress conditions, with the maximum principal stress ($\sigma_1$)
acting in the axial direction. The radial principal stresses ($\sigma_2$ and $\sigma_3$) were generated by the confining pressure, i.e.,
$\sigma_2 = \sigma_3 = P_c$. The effective pressure is calculated as $P_{eff} = P_c - P_f$. During deformation, a constant pressure difference



of 0.5 MPa was maintained between the inlet and outlet of the pore pressure system, while the mean pore pressure
was kept at 5 MPa. We thus maintained fluid flow across the sample and measured the permeability evolution during
deformation. In one $H_2O+CO_2$ experiment (OR2_M), we closed the fluid mixing vessel after the initial filling of the
sample and thus formed a close pore fluid loop (OR2_M was referred to as $H_2O+CO_2$ close experiment in the following
discussion). In the other $H_2O+CO_2$ experiment (OR3_B), the pore fluid system was connected to the fluid mixing
vessel during the entire experiment and therefore acted as a semi-open system since it was in constant communication
with a large $CO_2$ source (OR3_B was referred to as $H_2O+CO_2$ open experiment in the following discussion).

| Experiment Number | Sample Length (mm) | Sample Diameter (mm) | Confining Pressure (MPa) | Pore Pressure (MPa) | Effective Pressure (MPa) | Pore Fluid Composition | Temperature (°C) | Young's Modulus (GPa) | Ultimate Strength (MPa) | Strain at Failure (%) | Initial Porosity (%) |
|---|---|---|---|---|---|---|---|---|---|---|---|
| OR5 | 77.37 | 39.32 | 50 | 0 | | - | | 17.6 | >105 | 1.89 | 15 |
| OR2_T | 81.5 | 38.01 | 55 | 5 | | $H_2O$ | | 12.1 | 72 | 1.71 | 11 |
| OR2_M | 81.48 | 39.22 | 55 | 5 | 50 | $H_2O + CO_2$ | 78 | 16.2 | 55 | 0.84 | 5 |
| OR3_B | 77.94 | 39.81 | 55 | 5 | | $H_2O + CO_2$ | | 28.0 | 130 | 2.00 | - |

**Table 1.** Details of the sample parameters and experimental conditions. Sample OR5 was not loaded to its ultimate
strength due to early failure of the strain gauges and LVDTs. Porosity is estimated from the X-ray tomographic image
of the sample. Initial porosity of the sample OR3_B is not available due to limited access to the X-ray tomography
facility during COVID-19 pandemic.
We started the experiments by bringing the sample to an effective pressure of 50 MPa and subsequently to a
temperature of ~80°C while holding the pressure constant. Heating the sample took ~12 hours, long enough to allow
thermal equilibrium to be reached. After reaching the desired P - T conditions, the samples were deformed using a
step loading procedure. During a step, the load was increased at a rate of ~2 MPa/min, which corresponds to an axial
strain rate of ~$1.1\times10^{-6}$ s$^{-1}$. Once the desired stress level was reached, we kept the load constant for ~24 hours, while
monitoring the sample deformation. This sequence was repeated as many times as desired for the next loading steps.
The total duration of the experiments ranged between 5 to 12 days. Details of the load steps are summarized in
Appendix (Figure A1).
In this study, we use the term phase I to refer to the creep immediately following a stress change, during which strain
evolves rapidly. We call phase II the portion of the creep curve with an approximately constant or very slowly varying
strain rate over a ~24h window (i.e. $d\varepsilon/dt = cte.$; see Appendix Figure A2). For comparison with previous work on
brittle creep, we calculate a characteristic creep strain rate using a least-squares fit to the slope of the creep strain vs.
time curve during the identified phase II transient creep (Appendix Figure A4; we will simply refer to it as creep rate
in the following discussion).
To investigate the micro-structural changes occurring during deformation, the rock samples were scanned before and
after deformation using X-ray computed tomography with scan parameters set at ~150 kV and ~250 μA. The obtained





X-ray images have a pixel size of ~90×90 μm. Thin sections were prepared from selected samples and imaged using
a field emission scanning electron microscope (SEM).
The evolution of fluid composition was evaluated by collecting fluid samples from the end of the pore fluid outlet
(Figure 2a) after each creep step. The concentration of $Mg^{2+}$, $Ca^{2+}$ in the fluid sample were analyzed using the
Inductively Coupled Plasma Mass Spectrometry (ICP-MS).
**3 Results**
**3.1 Creep Deformation and Creep Strain Rate**
The creep deformation during each load step exhibited typical transient creep evolution (Brantut et al., 2013;
Robertson, 1964; Scholz, 1968) with a transition from rapidly evolving phase I to slowly varying phase II (Figure 3).
This transition generally took place within the first $10^4$ s (~2.7 hrs) of the loading step. In the dry experiment, large
variations in phase I creep strains were observed (Figure 3a & 4c) and the creep rates measured during the slowly
evolving phase II stages showed a weakly negative sensitivity to stress (Figure 3e). In experiments where pore fluids
were present ($H_2O$ and $H_2O+CO_2$), the strain accumulated during the phase I creep systematically increased with
increasing stress and the creep strain rate displayed a clear exponential dependence on stress (Figure 3e). This stress
sensitivity of creep strain rate showed strong similarity in the different experiments irrespective of the pore fluid
composition and can be adequately described by power law (e.g. Atkinson, 1984; Meredith and Atkinson, 1983) as
well as exponential functionals (Charles and Hillig, 1962; Hillig, 2006), but the exponential model seems to work
slightly better with our data according to the $R^2$ value (see Appendix Figure A5).


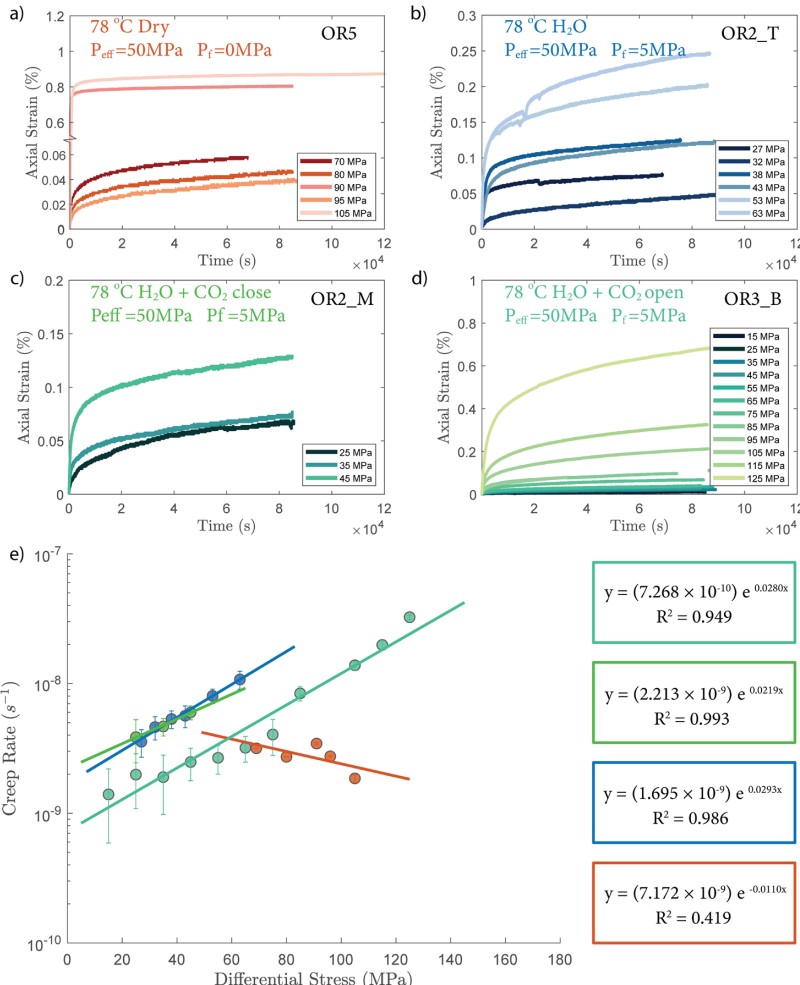

**Figure 3**. a) - d) Axial strain evolution of each individual stress steps for all experiments; e) Stress dependence of creep rate. The error bar marks the 95% confidence interval of the calculated creep rates. The stress-creep rate relationship can be best modeled using an exponential law.

In Figure 4 we compare the strain accumulation during phase I and II of the transient creep as illustrated in Figure A4a. We observe a universal power-law relationship of the accumulated creep strain during phase I to that measured at an arbitrary observation time of ~24 hrs after the stress step loading during phase II, in all experiments irrespective of fluid presence or the composition of the fluid (Figure 4a). The accumulated creep strains during both phase I and phase II were exponentially dependent on creep stress (Figure 4c and 4d). In Figure 4b we show that regardless of the creep stress level, the ratio between the logarithmic accumulated phase I and logarithmic phase II creep strain after ~24 hrs was approximately constant, except for two outliers associated with two stress steps in the dry experiment, during which anomalously large phase I creep strains occurred (Figure 4a and 4c).





Overall, the fluid saturated samples crept faster than the dry sample during phase II stages in similar stress conditions.
In spite of variations in ultimate strength, the fluid saturated samples consistently showed stronger stress dependence
of the creep rate than the dry sample. Comparing the fluid saturated experiments, we observe that the sample saturated
with $H_2O$ had the same creep rate as the $H_2O + CO_2$ close experiment and a higher creep rate than the $H_2O+CO_2$ open
experiment under similar stress level. (Figure 3e). Analysis of the fluid chemistry demonstrates that the $H_2O + CO_2$
close and $H_2O$ experiment show same fluid composition which we will describe in more detail in Section 3.6.

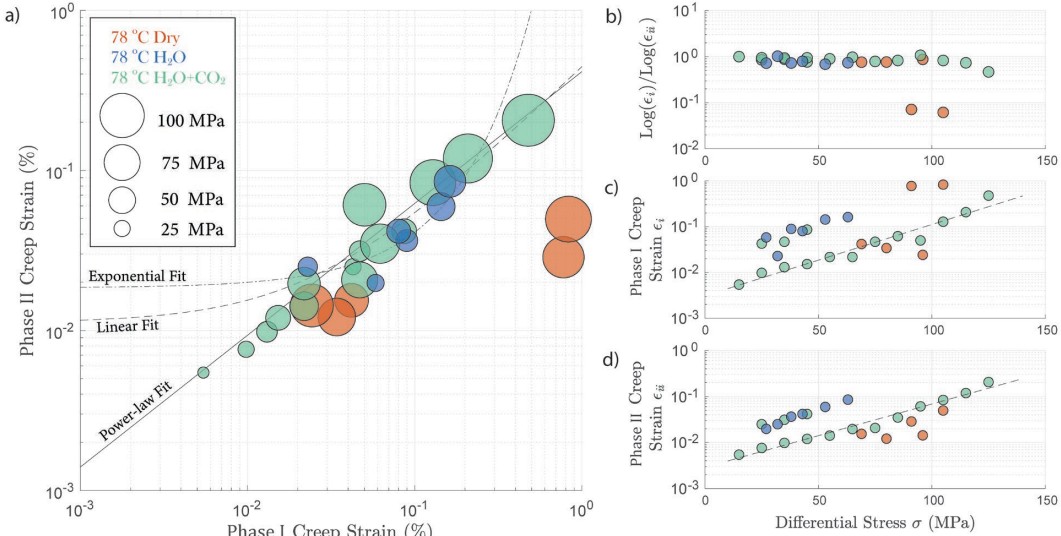


**Figure 4**. a) Relationship between total phase I creep strain and phase II creep strain ~24 hrs after the stress step
loading. The creep stress level is reflected by the size of the circles; b) ratio between the logarithmic total phase I and
logarithmic phase II creep strain remains constant and is independent of stress; the cumulated c) phase I and d) phase
II creep strain is exponentially dependent on the creep stress; The fitted lines are calculated based on the data obtained
from $H_2O + CO_2$ open experiment (OR3_M).

**3.2 Volumetric Strain**

In all experiments, creep deformation was initially compressive as indicated by a positive change in the volumetric
strain, $\epsilon_v$, calculated from the strain gauge measurements ($\epsilon_v = \epsilon_a + 2\epsilon_r$). Shear-enhanced dilation (Brace et al., 1966)
started 10 - 20 MPa before the ultimate strength of the sample was reached (highlighted by yellow arrowheads in
Figure 5). The onset of dilation generally occurred at lower stress level in the fluid saturated experiments than in dry
conditions. The largest dilation was observed in $H_2O+CO_2$ open experiments as shown in Figure 5d. In the dry
experiment, large amount of dilation ($\Delta\epsilon_v > 0.5\%$) was also observed at creep stress of ~90 MPa and ~105 MPa which
is significantly higher than in other steps ($\Delta\epsilon_v < 0.1\%$). Furthermore, the dilation at ~90 MPa is also accompanied by
a drop in stress (see Figure 5). The strength of the tested samples seems to be correlated with the elastic modulus
measurements, the stiffer the rock the higher the strength (see Table 1).

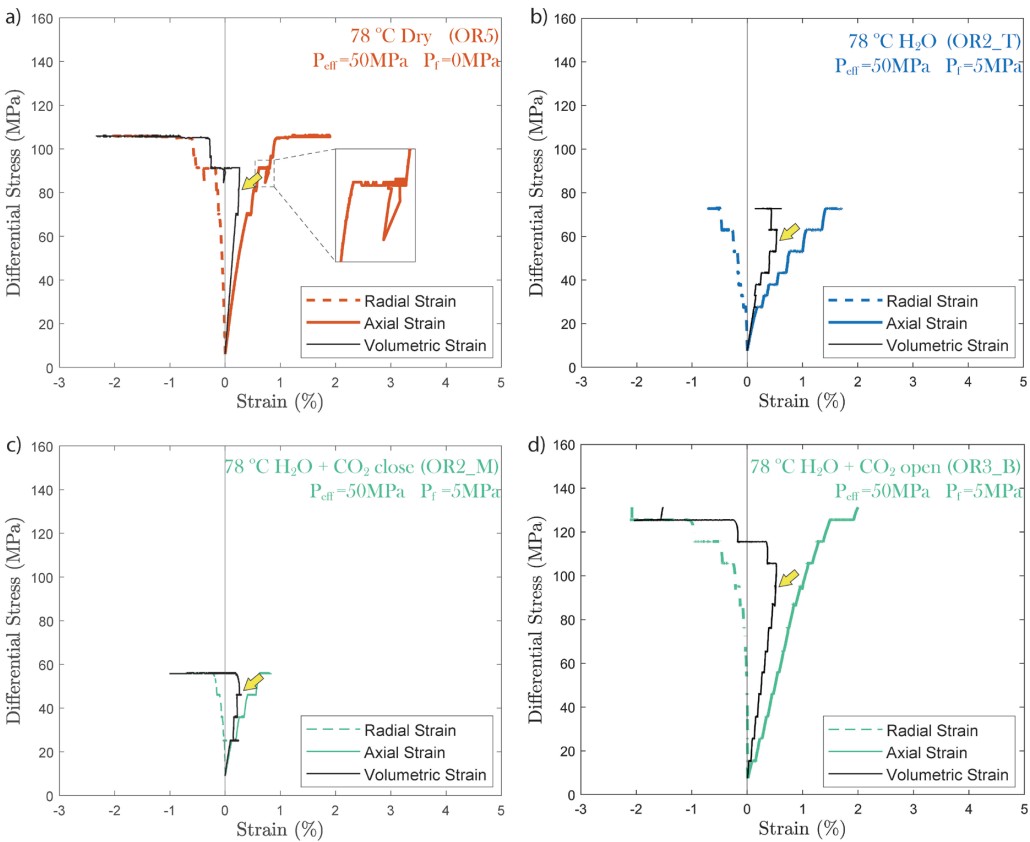

**Figure 5.** Plots of volumetric strain for a) dry, b) $H_2O$, c) $H_2O+CO_2$ close and d) $H_2O+CO_2$ open experiments. The onset of dilatancy is marked by the yellow arrowhead. In the dry experiment, the differential stress exhibits temporary fluctuation at ~90 MPa (highlighted by the dashed rectangle).

**3.3 Permeability**

In fluid-saturated experiments, permeability decreased with increasing effective pressure during hydrostatic loading (Figure 6a, b and c). The largest decrease in permeability was observed in the water-saturated experiment, where permeability dropped by 3 orders of magnitude as effective pressure was raised from 15 to 50 MPa (Figure 6a). Permeability reduction was much lower in both $H_2O+CO_2$ experiments, only ~ 1 order of magnitude, over the same effective pressure range (Figure 6bc). Permeability variations after heating are shown in Figure 6d 6e and 6f, where the minimum permeability reached during hydrostatic loading is indicated for comparison (empty circles in Figure 6d, e and f). The permeability change during heating was rather small in the $H_2O$ and $H_2O+CO_2$ close experiment, while the $H_2O+CO_2$ open experiment exhibited more than an order of magnitude permeability reduction after heating.

During creep, permeability did not evolve much with time but did show a clear dependence with the stress level of the individual creep stages, first slightly decreasing with increasing differential stress and then substantially increasing when the onset of shear-enhanced dilation was passed, shortly before failure (Figure 6d and 6f).


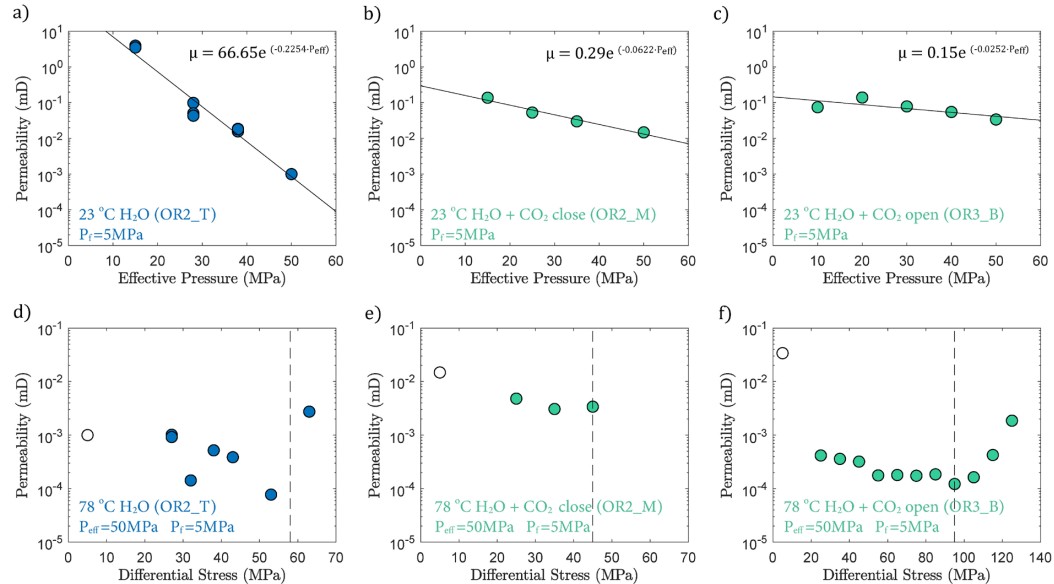

248

**Figure 6**. Permeability evolution due to changes in a) b) c) effective pressure and d) e) f) differential stress. The dash-line marks the onset of shear-enhanced dilation as previously shown in Figure 4. The empty circle indicates the permeability measurement before heating.

## 3.4 Characterization of the Acoustic Emissions

### 3.4.1 Passive Recording & Rate of AEs

We observed a strong correlation between acoustic emissions and mechanical data as documented in Figure 7. The number and amplitude of AEs was substantially larger in the experiments with pore fluids than in the dry experiment, irrespective of fluid composition. The rate of AEs increased during primary creep; the greater the accommodated strain was, the higher the AE rate. The AE rate then decayed exponentially as the rock entered the later stage of the transient creep. This decay was slower in all fluid saturated experiments where significant amount of AE activity continued during the phase II creep stage. The AE rate increased as the stress was approaching the ultimate strength of the sample (Figure 7). In Figure 8, we plot the normalized cumulative AE counts against the normalized creep strain measured during each creep step. For all experiments with pore fluids, we see that the data-points tended to cluster near the 0-1 diagonal (Figure 8b, c and d), thus supporting a strong correlation between acoustic emissions and creep strain. In the dry experiment, most AEs occurred early in each load step (normalized strain $\leq 0.2$) after which straining continued with little AE activity (Figure 8a).


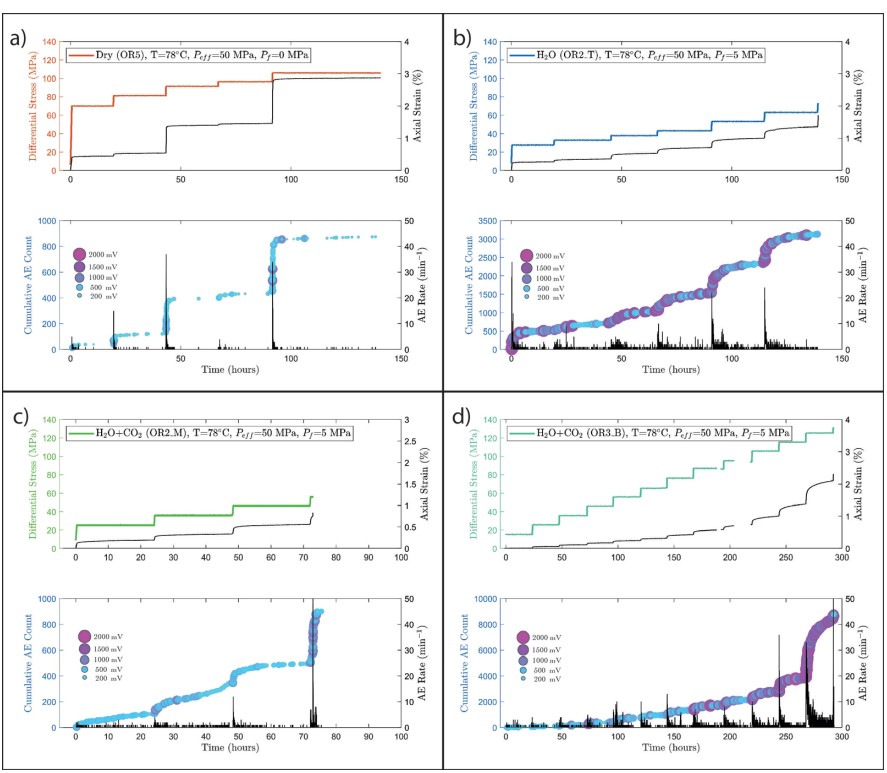

**Figure 7**. Top: Plot of stress loading steps and strain (black) evolution; Bottom: Evolution of cumulative number of
acoustic emission (AE) and AE rate evolution (black) over time for a) dry, b) $H_2O$, c) $H_2O+CO_2$ close and d) $H_2O+CO_2$
open experiments.

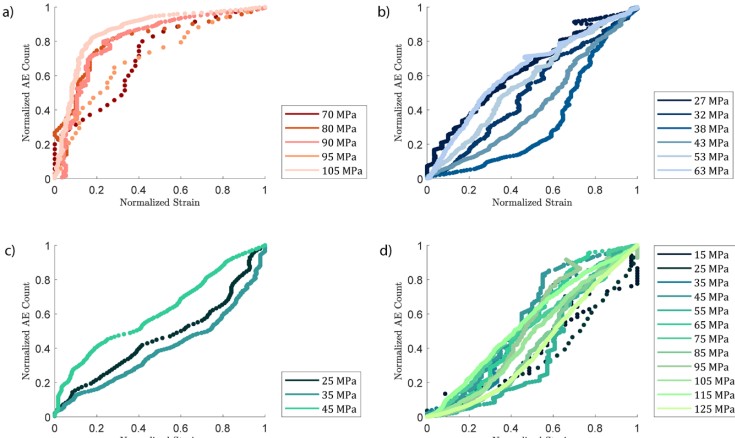


**Figure 8**. Plot of normalized cumulative AE count vs strain a) dry, b) $H_2O$, c) $H_2O+CO_2$ close and d) $H_2O+CO_2$ open
experiments. the normalized cumulative AE counts and strain during each creep step show an approximately linear
correlation in most cases except in the dry experiment.





### 3.4.2 Gutenberg-Richter b-value


The Gutenberg-Richter relationship provides a way to characterize the AE statistics for each creep step. The
Gutenberg-Richter $b$-value was calculated using the following equation:
$$\log N = a - b \log A \tag{Eq. 7}$$
where $A$ is the maximal amplitude of individual acoustic events and $N$ is the number of events with magnitude larger
than $A$. Figure 9 shows that the $b$-value increased with increasing stress in the fluid saturated experiments but remained
constant in the dry experiment. The observed increases of the $b$-values indicate that low amplitude AEs had a
proportionally larger frequency with increasing stress.

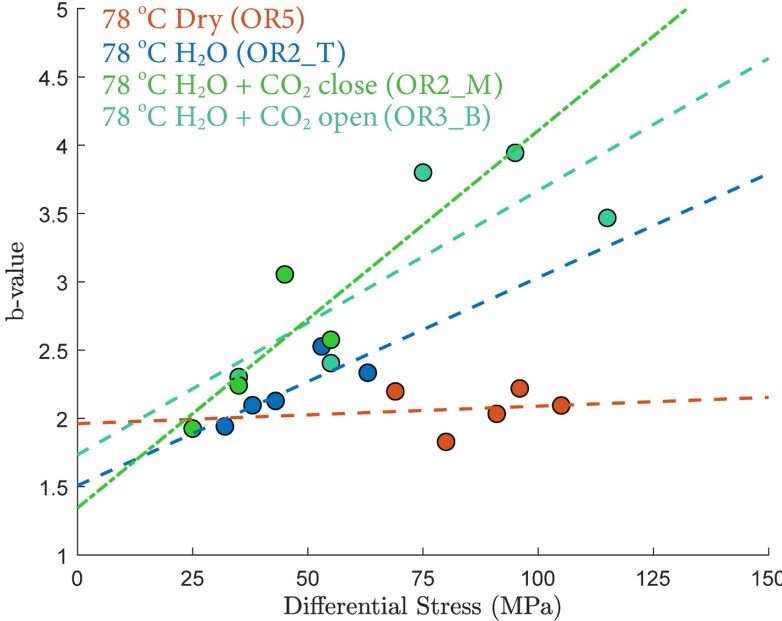


**Figure 9**. Differential stress dependence of Gutenberg-Richter $b$-values. See detailed b-value fitting in the Appendix
Figure A6.

### 3.5 Microstructure


Post-mortem examination of the samples reveals that fractures inside the fluid-saturated samples form a complex,
wide system rather than a clearly defined, distinct shear fault plane (Figure 10 and Appendix Figure A7). The fluid-
saturated samples exhibit bulging on the surface. In contrast, the dry sample shows a weakly developed fault plane
and less bulging, however it should be noted that this sample did not, in fact, reach ultimate strength.





$H_2O + CO_2$ (OR3_B)

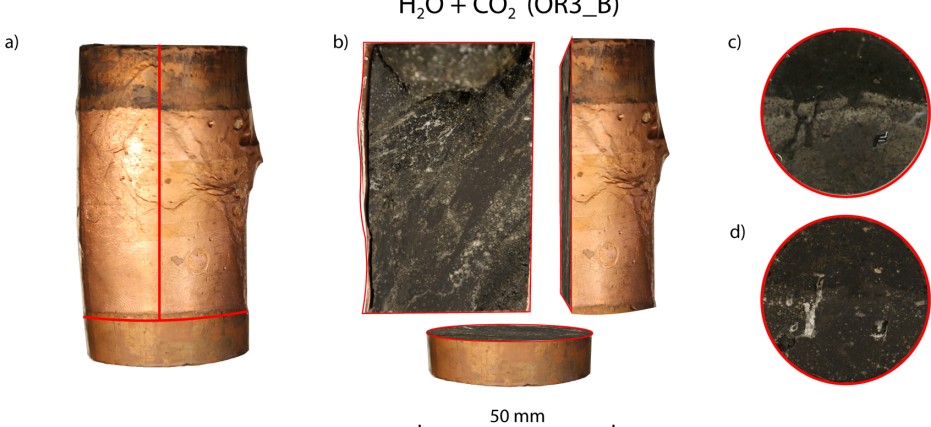

**Figure 10**. a) Deformed sample from $H_2O+CO_2$ open experiment (OR3_B). b) Cross-sectional view of the deformed
sample. Top view of the sample c) before and d) after the experiment.
X-ray tomographic images (Figure 11) and BSE images (Figure 12) of the deformed samples display abundant
fractures, whereas cracks are much more rare in the pre-deformation CT scans and the BSE images (Figure 1). The
amount of visible cracks in each sample tends to scale with the cumulative AE count; the dry experiment has a lower
fracture density than the experiments with $H_2O$ and $H_2O + CO_2$ despite the fact that the dry sample experienced a
higher stress and developed a larger total strain accumulation. To illustrate these observations, we selected
representative pairs of 2D tomographic slices oriented parallel and perpendicular to the loading direction and traced
the observable microcracks (Figure 11). We quantified both the orientation and anisotropy of the microcracks using
the 'surfor' method that relies on the projection of an outline (Heilbronner and Barrett, 2014; Panozzo, 1984). As
documented in Figure 11, cracks are strongly aligned in the axial sections. The cracks are mainly oriented parallel to
the maximum principal stress in the $H_2O+CO_2$ experiment, indicating Mode I cracking, but are aligned 20~30° to the
maximum principal stress in the dry and $H_2O$ experiments, suggesting mixed Mode I + Mode II cracking. A weaker
alignment is generally observed in radial sections.
**3.6 Fluid Chemistry**
Concentration of the $Mg^{2+}$ and $Ca^{2+}$ cations increased once heating started (Figure 13). This increase in the $Mg^{2+}$ and
$Ca^{2+}$ concentration reflects the dissolution of Mg and Ca bearing minerals during the reaction. In the $H_2O+CO_2$ close
experiment (OR2_M), the supply of $CO_2$ was limited and led to a dissolution dominated system that resulted in the
high concentration of $Mg^{2+}$ and $Ca^{2+}$, similar to the $H_2O$ experiment (OR2_T). In the $H_2O+CO_2$ open experiment
(OR3_B), the cation concentration was significantly lower than in the OR2_M and OR2_T experiments. This was
likely caused by the potential precipitation uptake owing to the continuous supply of $CO_2$ in the semi-open setting of
the pore fluid system. This interpretation is also supported by the ~2 orders of magnitude drop in permeability observed
in the $CO_2$ open experiment after heating started since precipitation could potentially clog the pore throats and lead to
permeability decrease.

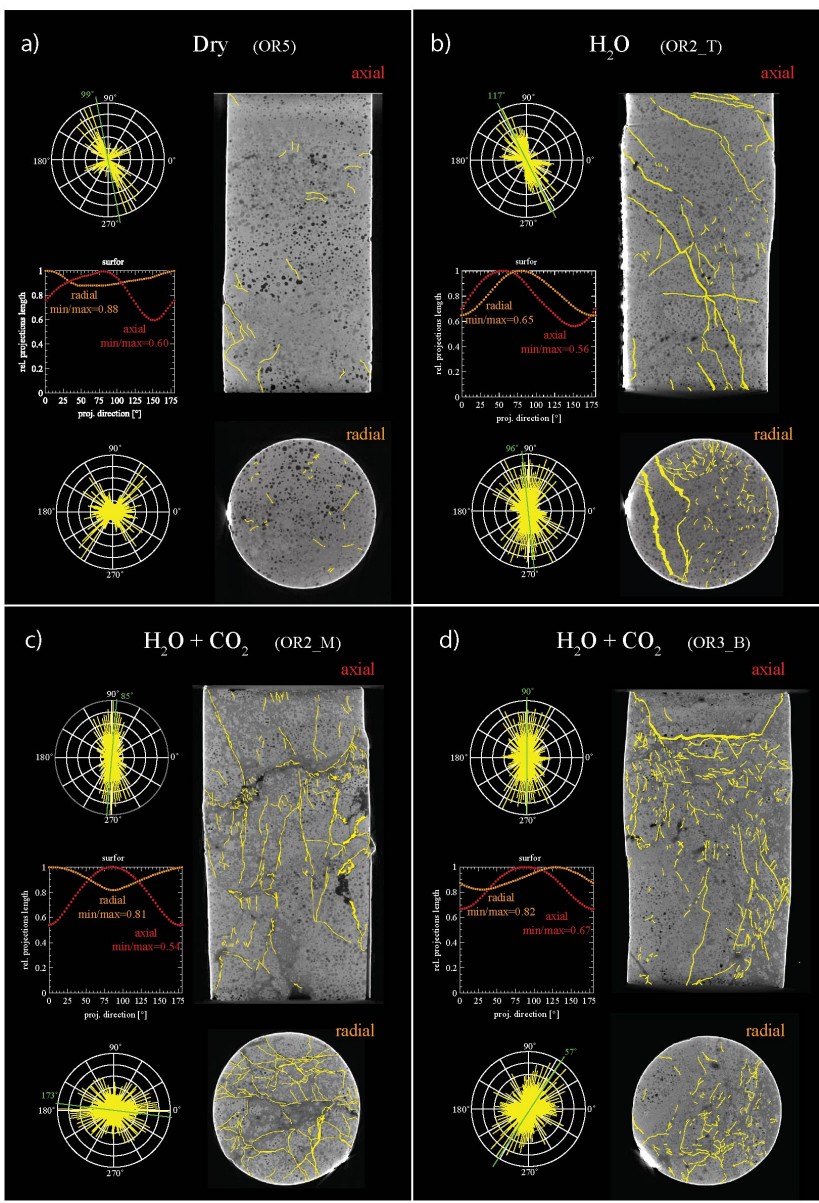


**Figure 11**. Orientation analysis of cracks in axial and radial directions from x-ray tomographic images of the deformed sample from a) dry, b) $H_2O$, c) $H_2O+CO_2$ close and d) $H_2O+CO_2$ open experiments. The aspect ratio (min/max), which is defined by the ratio between the minimum and maximum projection length of the cracks, is 1 when the orientation is random (isotropic) (Heilbronner and Barrett, 2014). Strong crack alignment is inferred in the axial sections with aspect ratio of 0.5~0.7 compared to the radial sections where the aspect ratio is 0.6~0.9.

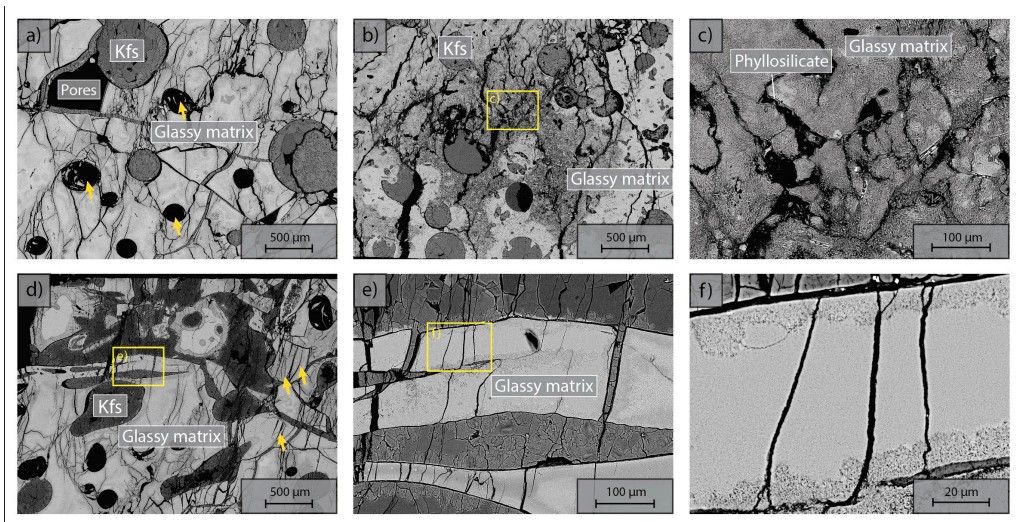

320
**Figure 12**. Microstructures of deformed sample from $H_2O+CO_2$ open experiment (OR3_B). Axial view, loading from
top and bottom. a) Pervasively fractured matrix, note the preferential N-S alignment of cracks indicating most cracks
are mode I. Note the collapse of void pores with cracks emanating (arrows). b) Crack pattern in altered glass matrix.
c) close up of b). Note the phyllosilicate coating on the crack wall. d - f) magnification cascade illustrating the crack
shape and morphology in the deformed sample.

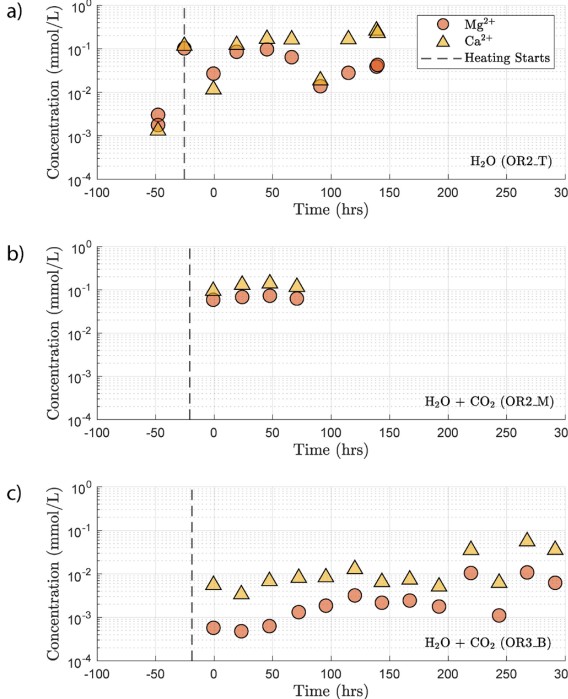

326
**Figure 13**. Concentration of $Mg^{2+}$ and $Ca^{2+}$ in the sampled fluid from a) $H_2O$, b) $H_2O+CO_2$ close and c) $H_2O+CO_2$
open experiments. Time 0 marks the start of the first creep stepping.



## 4. Discussion

### 4.1 Active Deformation Mechanisms

Acoustic emission, microstructure analysis and mechanical data confirm that the observed deformation is a brittle process as is expected at the given P-T conditions. The strong similarity between the time evolution of cumulative AE counts and strain (Figure 7 and 8) is consistent with observations from other creep deformation experiments using cemented and uncemented porous rocks (e.g. Brzesowsky et al., 2014; Heap et al., 2009). These considerations suggest that the creep deformation observed in this study is a result of a time-dependent brittle process such as subcritical cracking, that can still generate AE activity (Chester et al., 2007; Chester et al., 2004).

Previous studies concluded that brittle creep is unlikely to occur below the onset of dilatant cracking (*C'*) that is expected at about 80% of ultimate failure strength (Baud & Meredith, 1997; Heap et al., 2009). However, all our strain measurements (strain gauges, LVDTs, axial ram displacement) show that, in this study, creep did occur at stress levels of only ~11% of ultimate failure strength, well below *C'*. Similar creep deformation with measurable strain at low stress level has been previously observed in shale (e.g. Mighani et al., 2019). It can be argued that such low-stress creep deformation is associated to shear-enhanced compaction instead of dilatant cracking and that a change of mechanism may take place at *C'* (Vajdova et al., 2012; Zhu et al., 2010). However, we found that the strain rates measured during the creep steps below *C'* could be fitted using the same exponential law derived from strain rate measurements above *C'*. Furthermore, the amount of creep strain accumulated during phase I and phase II showed a consistent stress dependence across all stress conditions (Figure 4c and 4d). Therefore, the creep deformation above and below the point of dilation (*C'*) was likely governed by the same mechanism, and the accumulated creep strain at a given time can be formulated as a function of stress.

Our AE statistics show that the *b*-values were higher for the fluid-saturated experiments than the dry experiment, indicating a higher proportion of low amplitude AEs (i.e., higher ratio of low-to-high amplitude events). This abundance of low amplitude events in fluid-saturated rock is a direct evidence that aqueous fluids promoted creep deformation in basalt. As argued in previous studies, growth of small cracks and low amplitude events are facilitated when stress corrosion is activated in the presence of aqueous fluids (Hatton et al., 1993). We also observed that the amplitude of the largest events increased with increasing stress. This could be attributed to the increase in micro-crack nucleation, consequently maximizing the likelihood of an 'avalanche' of coalescing cracks, which, in turn, generates large amplitude events. Overall, as more and more energy is dissipated through micro-cracking and the associated low amplitude AEs, the macroscopic deformation becomes less dynamic, which is consistent with the increase in the Gutenberg-Richter *b*-value with increasing stress.

Post-mortem examination of the fluid-saturated samples demonstrated the presence of a complicated network of fractures within the sample and absence of a well-defined major shear fracture plane. The samples also exhibited dilation features such as bulging, likely caused by the bulk formation of dilation cracks. These microstructural observations further support the idea that deformation during creep is diffuse and distributed rather than localized (Hatton et al., 1993; Heap et al., 2009), consistent with nucleation-controlled crack growth since the nucleation sites are normally randomly distributed in the sample.



Microstructure analysis of the deformed samples demonstrates that the presence of fluid resulted in more abundant
mode I cracks (Figure 11). Larger amount of cracks oriented parallel to the maximum principal stress were observed
in the $H_2O+CO_2$ samples, implying dominant Mode I cracking, while the dry experiment showed less cracking, with
the cracks aligned 20-30˚ to the maximum principal stress, thus pointing to mixed Mode I + Mode II cracking. This
observation is consistent with previous studies on strain localization as they often proposed rock fracture models
predicting that mode II cracking takes place during the localization stage of fracture development (Lockner et al.,
1992; Reches and Lockner, 1994; Wong and Einstein, 2009). Among the present experiments, the samples subjected
to creep deformation under $H_2O+CO_2$ conditions exhibited the largest amount of mode I cracks. The sample deformed
under dry condition, despite having experienced similar differential stress and  total accumulated strain, showed a
lower amount of cracks. As stated in previous studies, mode II cracks often propagate at velocities close to the
Rayleigh velocity, which increases the probability of occurrence of high amplitude events. On the other hand, mode I
cracks have significantly lower rupture velocities and tend to produce low amplitude acoustic events (Broberg, 2006).
Therefore, increased mode I cracking should lead to an increase in the proportion of low amplitude AEs, i.e. an
increase in the Gutenberg-Richter *b*-value.
We infer that the difference in creep rate of the dry and fluid-saturated experiments is a result of fluid-assisted
subcritical crack growth. The fluid presence promotes stress corrosion, possibly related to hydrolytic weakening
(Atkinson, 1984), accelerates crack growth, activates more crack nucleation sites, and, consequently, leads to a
distributed array of small micro-cracks. In contrast, crack growth under dry conditions is concentrated on fewer and
larger cracks since activation of the nucleation sites is more difficult. Thus, it is easier to create localized deformation
under dry condition.
Previous studies also suggested that intergranular pressure solution (IPS) could play a significant role as a deformation
mechanism during creep (Liteanu et al., 2012; Zhang & Spiers, 2005; Zhang et al., 2010). The creep deformation by
IPS involves dissolution and the presence of a fluid phase might be expected to affect creep deformation, generating
additional strain accumulation apart from dilatant cracking. Importantly, because the driving process of IPS is not
producing abrupt stress drops, it is not expected to produce acoustic emissions. Although we did see difference in
creep strain between the dry and fluid-saturated experiments, it was likely caused by dynamic fracturing, as evidenced
by the volumetric strain and AE observations (Figure 5a and A1).  We attribute the change in creep strain rate between
dry and fluid-saturated experiments to fluid-assisted subcritical crack growth. We posit that under our experimental
conditions, IPS was not a dominant creep mechanism, however more detailed microstructural observations are needed.
**4.2 Time and Stress Dependent Deformation**
Our experiments show that the time-dependent creep deformation was also strongly stress dependent. We observed
that the creep strain accumulated during phase I was exponentially dependent on stress (Figure 3c). Two exceptions
are noted in the dry experiment. Both showed high strain accumulation during phase I transient creep and followed a
sharp temporary stress drop during the creep step with a nominal differential stress of ~90 MPa (Figure 5). This
temporary stress drop was accompanied by a swarm of large amplitude AEs (Appendix Figure A1), implying that the



concurrent strong dilation was likely caused by local dynamic fracturing while the bulk of the sample remained mostly
intact and still capable of supporting the applied load.
We also observed an exponential relationship between stress and creep rate. Interestingly, the fluid-saturated
experiments yielded approximately equal stress sensitivities of the creep rates, $\dot{\varepsilon} \propto e^{0.02 \sim 0.03\,\sigma}$, despite the variability
in their absolute strengths (Figure 3). The exponential stress dependence of strain rate in fluid-saturated experiments
is consistent with brittle creep being the dominant deformation mechanism. Indeed, the values of the fitting constant
(0.02~0.03) are comparable in order of magnitude to those reported in previous studies on other basaltic rocks (0.05
in Heap et al., 2011, from experiments using Etna Basalt). Since the creep rate was exponentially dependent on stress,
so should be the accumulated phase II creep strain. This inference is supported by our observation in Figure 4d.
Concerning the dry experiment, we attribute the slightly negative dependence of creep strain rate on stress (Figure 3e)
to damage-related strain hardening. However, it is also possible that this observed negative dependence was only a
statistical artefact owing to large data fluctuations as suggested by the low $R^2$ value of the exponential fitting
(Appendix Figure A5).
The fact that both cumulative phase I and phase II creep strains were exponentially dependent on stress (Figure 4c and
4d) implies a power-law relationship between the accumulated phase I and phase II creep strain, which is independent
of the stress level and even the presence or absence of fluids. This empirical relationship can be formulated as:
$$\frac{\log (\varepsilon_t - \varepsilon_i)}{\log (\varepsilon_i)} = \frac{\log (\varepsilon_{ii})}{\log (\varepsilon_i)} = cte. \qquad (Eq.\,8)$$

where $\varepsilon_t$ is the total strain accumulated at the end of an individual creep stage (~24 hrs), $\varepsilon_i$ the creep strain accumulated
during phase I, and $\varepsilon_{ii}$ the strain accumulated during phase II (see Appendix Figure A3). This phenomenological
power-law relationship is supported by our observation that the ratios in Equation 8 were indeed approximately
constant ~0.8 (Figure 4a and 4b). This power-law relationship expressed in Equation 8 implies that the strain evolution
with time can be predicted; some fundamental link between strain accommodated in phase I creep and strain rate in
phase II creep exists.
**4.3 Fluid Chemistry Evolution and Influence of Fluid Composition**
The increase in concentration of both $Mg^{2+}$ and $Ca^{2+}$ occurring after heating in the $H_2O$ and $H_2O+CO_2$ close experiment
(Figure 14) indicates that the system was dominated by dissolution of Mg and Ca bearing minerals. In the case of the
$H_2O+CO_2$ open experiment, we observed a much smaller increase in cation concentration implying that a significant
amount of the released $Mg^{2+}$ and $Ca^{2+}$ cations reacted with the continuously supplied $CO_2$ in the semi-open setting to
form carbonate precipitates. These cation concentration trends appeared strongly correlated with the permeability
evolution and creep strength of the rocks. The experiment with $H_2O+CO_2$ open showed a larger post-heating
permeability decrease than the experiments with $H_2O$ and $H_2O + CO_2$ close and was stronger (Figures 3e and 6).The
absolute creep rate was consistent for experiments with comparable fluid chemistry ($H_2O$ and $H_2O + CO_2$ closed) and
about a factor of 3 faster than in the experiment where precipitation was dominant ($H_2O + CO_2$ open) indicating that
precipitation reactions slightly strengthen the rock. This congruence of observations is a strong argument that
precipitation occurred in the pore space of the $CO_2$ open experiment. However, we could not directly resolve evidence





of precipitation within the resolution of our microstructural observations and this requires further study. Interestingly,
the strain rate sensitivity to stress was similar in all fluid-saturated experiments (Figure 3), implying that creep rate
sensitivity to stress was not significantly influenced by the fluid chemistry.
Our chemical data support the idea that carbonation of basalt is a kinetically favored reaction and are consistent with
the fast rate of carbonation observed during the CarbFix field tests (Matter et al., 2016). The difference between the
$Mg^{2+}$ and $Ca^{2+}$ concentrations measured in the $H_2O+CO_2$ open experiment and those in the $H_2O$ and $H_2O+CO_2$ close
experiments indicates that the rate-limiting factor during carbonation under our experimental condition was the net
supply of $Mg^{2+}$ and $Ca^{2+}$ cations, which is associated with dissolution and is reduced when precipitation occurs.

### 4.4 Permeability & Porosity Evolution

Permeability was affected by both chemical and mechanical processes. The evolution of permeability during the
experiments was generally consistent with previous observations of monotonic permeability decrease during
hydrostatic loading of samples of limestone, sandstone and Etna basalt (Brantut, 2015; Fortin et al., 2011; Zhu &
Wong, 1997). Comparison of the dissolution dominated experiments ($H_2O$ and $H_2O+CO_2$ close) and the precipitation
dominated experiment ($H_2O+CO_2$ open) shows that the carbonation reaction reduced permeability in our experiment.
In low differential stress conditions, the samples compacted and this compaction was accompanied by a further
permeability decrease, which was likely related to the pore volume reduction expected during compressive
deformation. Shortly before ultimate strength was reached, volumetric dilation became dominant and coincided with
permeability increase. Our observations of the permeability evolution demonstrate that, although the permeability
might decrease owing to compaction, formation and propagation of cracks can mitigate the permeability loss and even
lead to an increase with further cracking. The effect of creep deformation on the long-term permeability evolution of
reservoir rocks is therefore non-negligible. Increase in permeability, combined with other observations such as
increasing volumetric strain and acoustic emissions, could potentially be used as a warning sign for impending failure
during the long-term monitoring of reservoirs' integrity in GCS applications.

### 4.5 Effect of Sample Heterogeneity

As our samples are taken from drill cores collected at depth at the CarbFix carbon mineralization site, the heterogeneity
is larger than in rocks typically used in rock mechanics experiments. The samples investigated in this study exhibit
variations in their initial porosity (5-15%, see Table 1), ultimate strength (55-130 MPa) and Young's modulus (12-28
GPa). We observed a correlation between the ultimate strength and the elastic modulus of the samples where stiffer
samples reach higher peak strengths, consistent with previous reports of an empirical relationship between the
unconfined compressive strength and the elastic modulus of sedimentary rocks (see review in Chang et al., 2006). The
peak strength however varied inversely with porosity;  the dry sample (OR5), which has the highest initial porosity
(15%), shows a higher ultimate strength (>105 MPa) and exhibits the lowest creep rate compared to the fluid-saturated
experiments where porosity measurements were available ($H_2O$ and $H_2O+CO_2$ close). Remarkably, the stress
sensitivity of the creep strain rate shows consistency ($e^{0.02 - 0.03\ \sigma}$) in all the fluid saturated experiments ($H_2O$ and
$H_2O+CO_2$ open and close) in spite of these variations in porosity, stiffness and ultimate strength. Moreover, the creep
rate at individual stress steps is consistent for experiments with comparable fluid chemistry ($H_2O$ and $H_2O + CO_2$





closed) despite a variation in porosity by a factor of 2 in between the samples (Figure 3 and Table 1). These results
are a strong argument for the operation of chemical processes that contribute to creep. While variations in porosity
resulted in variation in peak strength, they did not seem to affect the absolute creep rates or the sensitivity of creep
rate to stress.

**5 Conclusions**

Through the experimental study of long-term creep deformation of Iceland Basalt, we have demonstrated that:
• Transient creep occurred at stress levels significantly below the onset of dilatant cracking.
• Presence of an aqueous pore fluid exerted first order control on the creep deformation of the basaltic rocks,
while the fluid composition had only a secondary effect under our experimental conditions. At similar
differential stress level, the creep rates in fluid-saturated experiments were much higher than the rates in the
dry experiment.
• A close system tended to favor dissolution over precipitation during carbonation in our experimental setting,
whereas precipitation played a more important role in an open system with continuous $CO_2$ supply.
• Larger amount of dilation was observed in fluid saturated experiments than in the dry experiment, as
evidenced by both volumetric strain data and micro-structural observations.
• Larger low/high amplitude ratios of the AE events and higher AE rates were observed during the phase II
creep of the fluid-saturated experiments than the dry experiment, indicating that aqueous fluids promoted
stress corrosion processes.
• The mechanism governing the creep deformation was brittle, time- and stress-dependent, and could likely be
identified as sub-critical dilatant cracking.
Overall, our results emphasize the non-negligible role that the creep deformation can potentially play in the long-term
deformation of rocks even under low pressure and temperature conditions and calls for more attention to time-
dependent processes such as sub-critical micro-cracking in GCS applications. Under our experimental conditions, the
creep deformation and the associated fracture development were affected by the presence of aqueous fluids, implying
that reactive fluids could potentially alter the fracture patterns and allow mineralization in a greater rock volume during
GCS applications. Further detailed studies on the creep deformation under chemically active environment are required
to better understand the long-term deformation of rocks in natural systems.

**Acknowledgments, Samples, and Data**

The authors benefited from discussions with Ben Holtzman, Yves Bernabé, Brian Evans, Bradford Hager and Brent
Minchew. The author would like to thank Yves Bernabé for his copy editing of this paper. The author would also like
to thank Edward Boyle and Richard Kayser for their help with the ICP-MS analysis. Funding by the MITei's Carbon
Capture, Utilization and Storage Center and NSF funding for CORD laboratory technician support (EAR-1833478
and EAR-2054414) are gratefully acknowledged. The x-ray tomographic images were obtained at the Center for
Nanoscale Systems (CNS), a member of the National Nanotechnology Coordinated Infrastructure Network (NNCI),
which is supported by the National Science Foundation under NSF award no. 1541959. CNS is part of Harvard
University. The cores used in this study were generously provided by Sandra Snæbjörnsdóttir and Kári Helgason. The
authors declare no conflict of interest. The underlying data is available at http:// doi.org/10.5281/zenodo.4926587

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



# Appendices

## A.1 Experimental Procedures

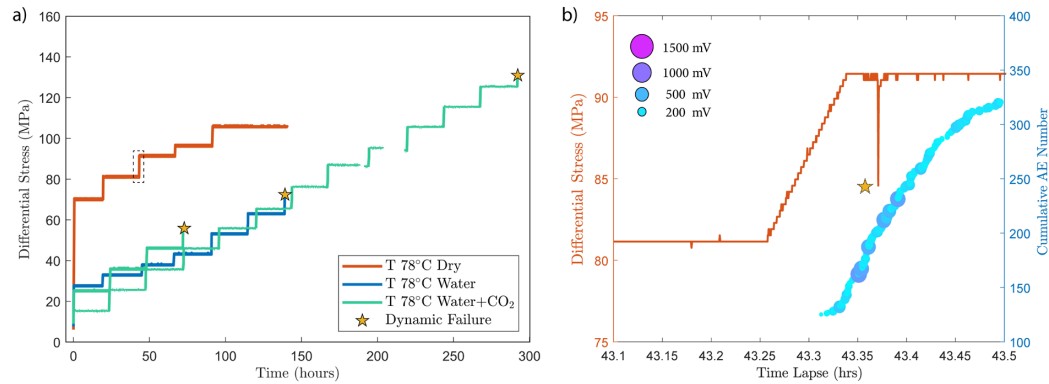

**Figure A1** a) Differential stress vs time plot of experiments conducted at temperature of 78°C. The dry experiment (red) was ceased before dynamic failure occurred in the sample. b) A temporary stress drop was observed (highlighted by the dashed rectangle in a)) during the primary creep of the dry experiment at creep stress of ~90 MPa accompanied by the occurrence of high amplitude AEs.

## A.2 Phase I to Phase II Transient Creep Transition

Selection of the phase II transient creep from the mechanical data is based on the calculated strain rate using first derivative of the strain curve vs. time at different stress levels (Figure A2). The plot of strain rate vs strain further supported that the strain rate evolution slows down during the identified phase II creep.

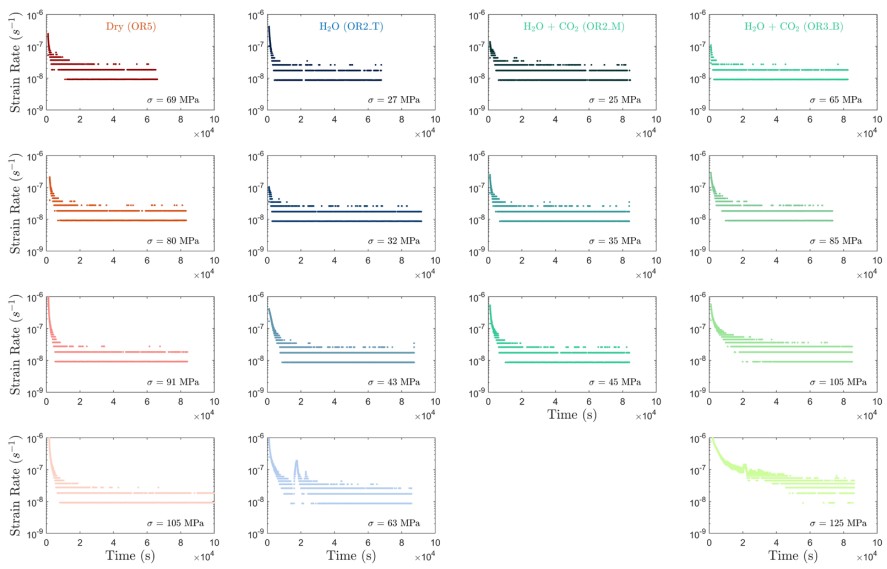





**Figure A2** Strain rate evolution calculated from the first derivative of the strain vs. time data. It can be observed that the strain rates generally become constant 10,000 s (≈2.8 h) after the load stepping in most steps.

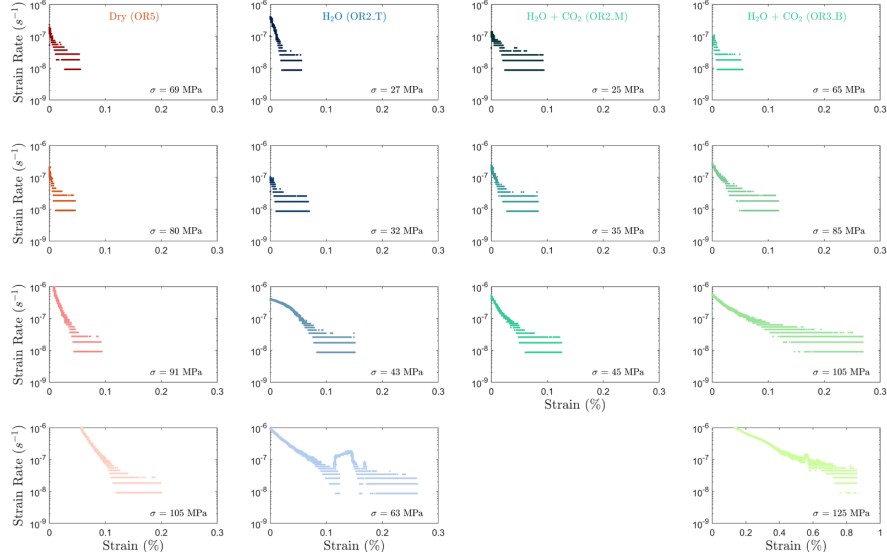

**Figure A3** Plot of strain rate evolution vs. strain.

To consistently analyze the transition between phase I and phase II of the transient creep, we fit the evolution of phase I creep strain over time using a power-law function and the phase II creep strain as a linear function (Figure A3 a). The measured strain data point that is the closest to the intersection of the two fitting functions is selected as the inflection point, i.e., the transition from phase I to phase II transient creep deformation. Figure A3 b and c shows the logarithmic and power-law fitting methods used for the time evolution of creep strain ($\varepsilon$). With the 24 hrs observation window of our experiment.

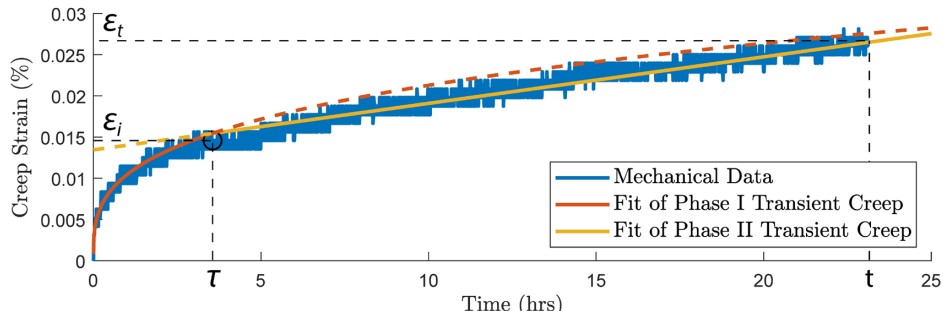

**Figure A4** illustration of the method used to pick up the transition (circle) from phase I to phase II transient creep deformation. a) Two phase model for the time evolution of creep strain. The transition (circle) from phase I to phase II creep deformation is selected based on the intersection of the power-law fit function of phase I creep (red) and linear fit function of the phase II creep (yellow).





**A.3 Creep Strain/Stress Models**

The strain rate during phase II creep deformation is generally described using the power law form (e.g. Atkinson, 1984; Meredith and Atkinson, 1983):

$$d\varepsilon/dt = A\sigma^n \qquad (\text{Eq. A1})$$

or the exponential form (e.g. Charles and Hillig, 1962):

$$d\varepsilon/dt = Be^{\eta\sigma} \qquad (\text{Eq. A2})$$

where $\varepsilon$ is the creep strain and $\sigma$ is the differential stress. $A$, $B$, $n$ and $\eta$ are constants. Both models have described our laboratory data well. The exponential model seems to be slightly better than the power law model when comparing the $R^2$ factors.

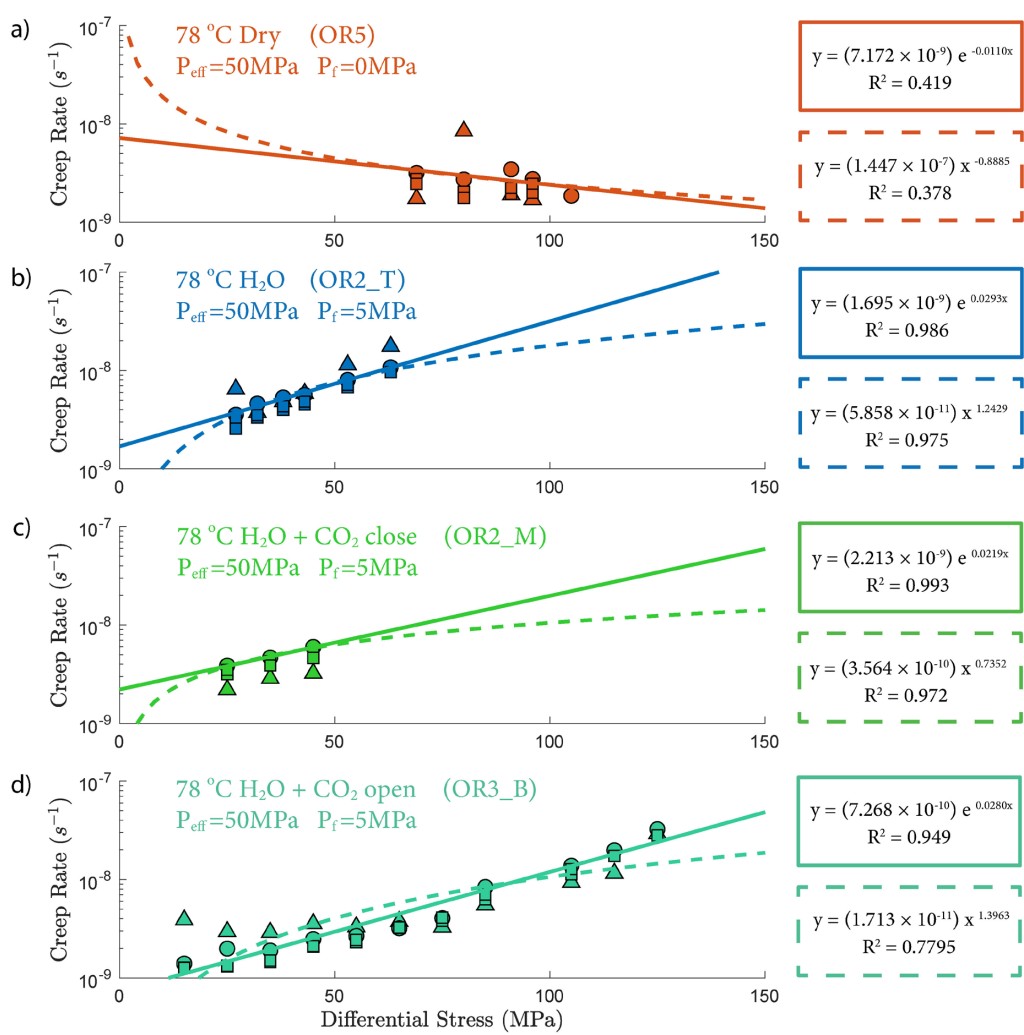

**Figure A5** Power-law (dash line) and exponential (solid line) fit of creep rate/stress relationship. The strain rates are calculated from strain measurement from main ram displacement (circle), strain gauge (triangle) and LVDTs (square).


**A.4 Gutenberg-Richter b-value**
Figure A6 shows the fitting of the Gutenberg-Richter b-value from different experiments at various stress levels.

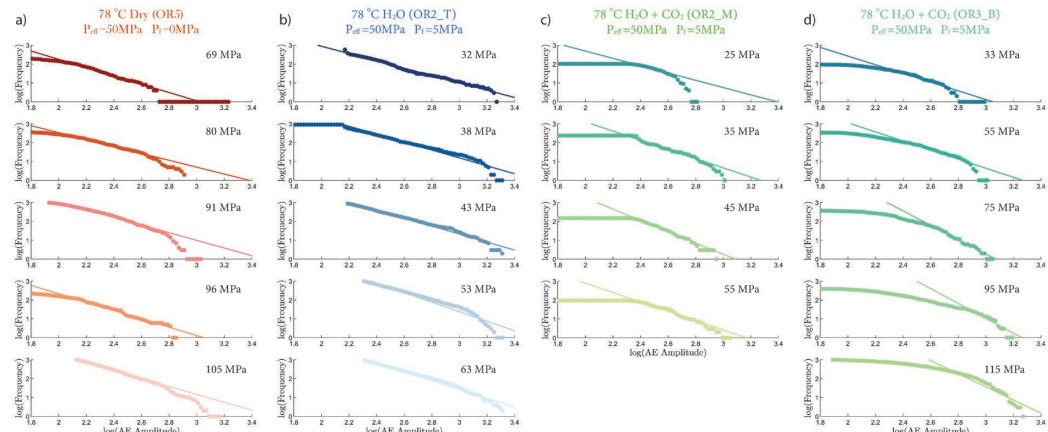

**Figure A6** Statistics of AE amplitudes for Gutenberg-Richter b-value calculation from a) dry, b) $H_2O$ c) $H_2O+CO_2$
close and d) $H_2O+CO_2$ open experiments.
**A.5 Samples after Deformation**

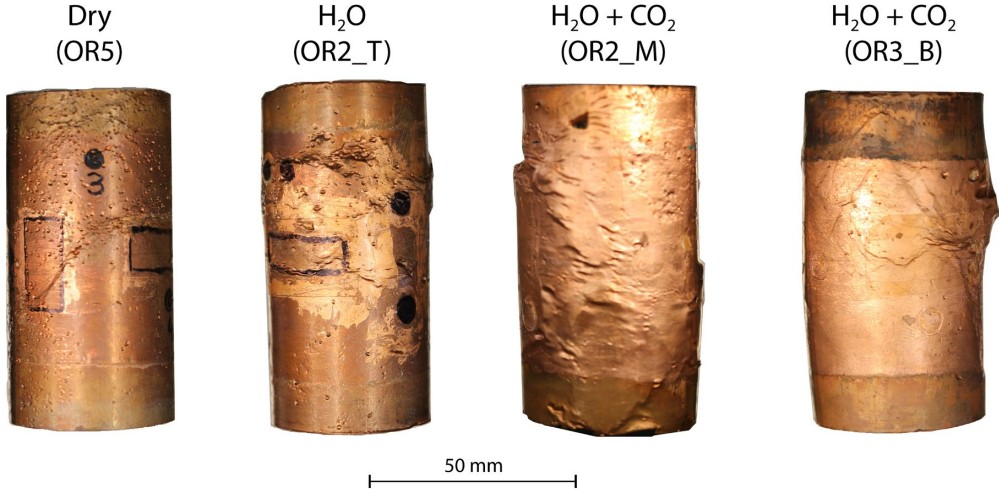

**Figure A7** Photo of samples after deformation. The dry sample did not reach the final dynamic failure before
experiment was halted.
**A.6 Elastic Modulus**
The Young's modulus ($E$) of the sample is calculated based on the strain measurement during the elastic loading,
using the following equation:
$$E = \frac{\Delta\sigma}{\Delta d/L}$$
(Eq. A4)

where $\Delta\sigma$ is the differential stress, $d$ is the displacement of main ram piston and $L$ is the length of the sample.

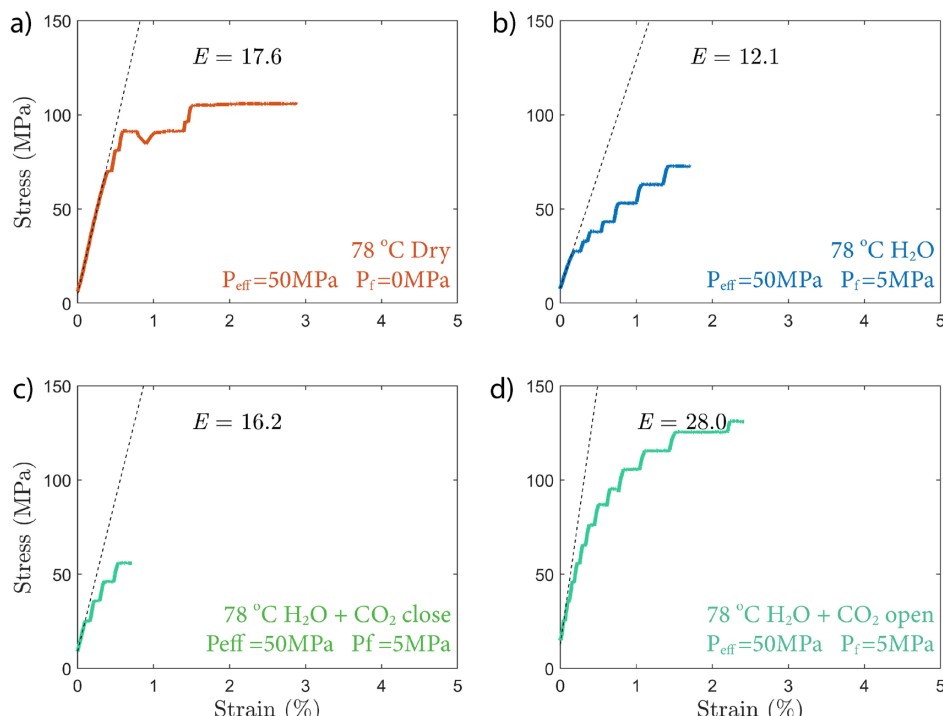


**Figure A8** Elastic modulus calculated from strain vs stress plots from a) dry, b) $H_2O$ c) $H_2O+CO_2$ close and d) $H_2O+CO_2$ open experiments.