# Peer review of "Creep of CarbFix Basalt: Influence of Rock-fluid Interaction"

_Solid Earth, 2021_

## Author Comment (AC1)

We thank Dr. Philip Benson for his constructive and detailed review. In this author comments letter, we addressed each comment separately. The reviewer's comments are italicized, followed by our point-by-point response to the reviewer.

*This paper presents an innovative suite of experiments investigating the link between creep in basalts in the presence of pore fluids and dissolved CO2. It is a novel contribution to the literature and pertinent to the study of CO2 sequestration within basalt via chemical (carbonation) reactions. The paper is well written, comprehensive in scope, and a pleasure to read. I have only a small number of minor, mostly technical, queries:*

- *Line 72: Maybe reference Heap (2011) here as well? (it is in the reference list) as I don't think creep in volcanic rock is a particularly common laboratory case study.*

We have added the reference here following the suggestion. (Line 71)

- *Line 167: Small edit required, Load is measured in kN, whereas MPa is stress, so I suggest a minor edit to used either one or the other.*

Thanks for pointing this out, we have revised the statement to 'the differential stress was increased at a rate of ~2 MPa/min'. (Line 177)

- *Line 199, figure 3 (and line 214). I panel (e) it would help me (and I hope the reader in general) with an annotation or two? Especially as some of those colours are similar – the fluid and conditions could usefully be added as a note on the plot.*

Thanks for the suggestion, we annotated the fitting lines with the experimental conditions. We kept the same color code for the plot in panel e) as panel a) to d) and added a statement in the caption of figure 3 to clarify this. (Line 216-220)

- *Section 3.3, in addition to the changes in permeability with effective pressure for the three samples (fig 6a-6c) can the authors say anything about the change in the initial permeability (at 0 MPa), which also seems to decrease as we move from 23C with just water, to water/CO2 (closed) and then to water/CO2 (open)?*

The variation in the initial permeability is likely caused by the sample-to-sample variation. We have conducted other experiment on the Iceland basalt which is not presented in this paper. And the initial permeability shows similar range of fluctuation.

- *Section 3.4.2 and figure 9: the seismic -b value data is interesting, but with only a few data points per experiment for fitting the line of best fit, I do have a concern regarding the scatter and fit (in figure 9). Do the authors have any sense for the error of the -b value trends presented here? If so, they ought to be discussed as there seems visually to be some degree of overlap between the experiments.*

The fitting of b-value is shown in Appendix Figure A7. We also added an error analysis in the fitting of the b-value to show the variations. The error bar shows the 95% confidence interval in the b-value fitting. (Line 306)

- *Line 425: The text references figure 14, but I think this refers to fig 13 on page 16?*

Thanks for pointing this out. We have corrected this in the revised manuscript.

---

## Author Comment (AC2)

We thank the anonymous reviewer for his/her constructive and detailed review. In this author comments letter, we addressed each comment separately. The reviewer's comments are italicized, followed by our point-by-point response to the reviewer.

*This is a review for "Creep of CarbFix Basalt: Influence of Rock-fluid Interaction" by Xing et al. This manuscript presents the results of deformation experiments on tholeiite samples from the CarbFix CO2 storage site under different pore fluid compositions. The goal of the study is to better understand the role of fluid composition, and CO2 in particular, on deformation of basalt. The authors used a stress stepping technique to evaluate brittle creep processes. Experiments that investigate fluid composition are few and far between, so this is a welcome study and approach. The rather large sample size used by the authors is also welcome.*

*I think that this data should be published, but I have several concerns with the analysis as described below. The results indicate that there are clearly differences in the mechanical behavior at different testing conditions, and I think the strongest link is between AE results, crack geometries, and stress-strain rate data. However, there are some issues with data interpretation that are detracting from this bigger picture in the present draft and some more detail on the chemistry of the system is needed.*

- *I think some of the confusion can be reduced by simplifying the analysis of the mechanical data to only what is critical to understanding the end goal. The problem also needs to be framed a bit more precisely which would help with this; while stress stepping experiments are very useful, they also provide a very specific kind of data and it is not clear why the authors chose these tests. I recommend presenting either a hypothesis or description of expected/anticipated results and the meaning of those results.*

We are interested in long-term mechanical behavior of the basalts in the presence of fluids, for this reason constant load experiments are much more appropriate than constant displacement rate experiments which typically last much shorter. The 'stress stepping' experimental procedure in this study is designed to minimize the issue caused by inter-sample variability and for collecting multiple stress - strain rate data points per sample. This method is also sample efficient considering the limited drill core availability. (Line 130-134) We have elaborated on the experimental procedure in section 2.2. (Line 180-185)

- *The rock composition is likely very important to the results presented and I think better compositional data is necessary to understand the results. First, the mineralogy is not clear from what is written. With the imaging tools available, it seems like it would be reasonable to estimate the modal mineralogy (also I think it should be mentioned if olivine is not present since it is involved in a potentially important reaction in some basalts). Beyond modal mineralogy, I strongly recommend using the resources available at MIT to determine the chemistry of the phases either qualitatively or quantitatively (what are the 'iron ore' and 'phyllosilicate phases?).*

We have revised the description on the sample composition. Previous studies from Alfredsson et al., (2013) and Larsson et al., (2002) shows that the primary minerals of the sample are predominantly plagioclase (An90-30), olivine (Fo90-80) and clinopyroxene (commonly augite), magnetite-ilmenite, and interstitial glass. (Line 101-105) We also added EDS data (Figure A1) showing the chemical composition of the starting material which show consistent result with previous studies.

*More detail about the experimental procedure is needed to really understand what was done and how best to analyze the results.*

- *What is the fluid composition in these experiments and how was is created/controlled? More information is needed about how the fluids are mixed and what the resulting composition is. For instance, what is the partial pressure of CO2? Is this the same during the open and closed experiments?*

We have clarified the text. The fluid is prepared in the fluid mixing vessel shown in Figure 2. Deionized water is saturated with $CO_2$ in the vessel under a gas pressure of 5 MPa. (Line 155-157)

- *I am unclear about why/how experiments were terminated. In at least some of the experiments it seems that the samples failed because the authors refer to 'ultimate strength', but that is not clearly explained and data demonstrating that is not shown (and no plot shown indicates anything that looks like failure to me). The dry test seems to have been terminated for other reasons, but that is not clear either. Understanding this would help to understand what is comparable between the samples.*

Most experiments were conducted until failure occurred and the sample lost its load-bearing capability. This typically occurred during a load step where the stress is changed rapidly. The dry experiment is halted earlier due to failure of the strain gauges and LVDTs. We added the statement in line 180-185 and in caption of Table 1.

- *Interpretation of the data.*

  1. *The authors refer to `ultimate strength' which they bever describe. Usually this term is used during constant strain rate tests. In these experiments, it is not clear what the significance of this value is, since failure in brittle creep is largely controlled by the amount of strain that is accommodated the rate of which is dependent on stress. If a sample fails at a lower stress when similar stress steps have been followed, as I think is being presented here, then that is probably a reflection of differences in the rate of processes, but that is not clearly explained or developed. The authors should clarify the meaning and significance of 'ultimate strength' and I recommend using a different term such as failure strength.*

We thank the reviewer for the suggestion and adopted the term 'failure strength' in referring to the stress level at which the sample failed in our experiments.

  2. *It looks like the authors are using the change from net compaction to net dilation to identify the onset of dilatancy (C') (Figure 5). This is not correct. The onset is identified as the diff stress at which the stress-vol strain curing deviates from the elastic hydrostat. Which is a lower stress that how it is identified by the authors. Without a hydrostatic loading curve it is very difficult to identify this transition. C' is generally thought to reflect the onset of microcracking, so it seems the authors have reached it in their experiments or they would have no results. Either way, they don't seem to have enough information to identify C' and the discussion in lines 337 to 348 need to be edited to reflect that.*

We thank the reviewer for point this out. We have changed the terminology here and are using the C*', which defines the transition from compactant to dilatant (Wong and Baud, 2012; also called D' in Brantut et al, 2012). Because of the stress-stepping method we adopted in this study, it is not straightforward to determine the onset of dilatancy (C') using the common method which marks the critical stress states when volumetric compaction decelerated in comparison with the hydrostat. Here, the C*' is selected based on the volumetric creep strain and is marked by the step where volumetric creep strain first exhibits dilation. We have modified our discussion accordingly.

- *The negative correlation between creep rate and differential stress in the dry test seems largely controlled by the highest differential stress. I recommend emphasizing that most of the data indicates stress-neutral behavior.*

We agree that the negative correlation between creep rate and differential stress is largely affected by the step 3 and 5 which is observed with dynamic failure processes. The lower $R^2$ value also indicates that the negative slope in the fitting is likely induced by the fluctuation of the data. We have modified the text to highlight the stress neutral behavior. (Line 218-220; Line 438-439)

- *Tertiary creep (the acceleration in strain that occurs before failure) is typically avoided as it does not have a steady state strain rate. How was that dealt with in these experiments? Is it possible that the high strain rates and AE rates at the highest stresses actually represent tertiary creep?*

We have clarified the text. In the present manuscript we focused on primary and secondary creep. The step where the ultimate failure occurs is omitted in the analysis of the stress/ strain rate relationship, i.e., we only include the steps where final failure is not observed (Line 187-188). We showed a typical step with developed tertiary creep in Figure 7c.

If the high strain rate and AE rates at the highest stresses represent tertiary creep, we would expect a different stress dependence of creep rate from the high stress steps. However, our observation shows that the exponential dependence is consistent throughout all stress steps. Therefore, we can rule out the possibility that high strain rate and AE rates at the high stresses represent tertiary creep.

- *I hesitate to ask for more experiments, but because of how the authors have chosen to interpret the data (with respect to C' and ultimate strength) hydrostatic and constant displacement rate tests under different conditions are an obvious way to clear up some of the issues. Another option is to avoid these concepts.*

We have clarified that we are using the transition from compactant to dilatant (C*') in our discussion and modified the statement accordingly.

- *There is almost no mention or description of carbonation products in these samples or their abundance. This is seems like a huge oversight given the goals of this project. There are before and after pictures in Figure 10, but no mention of what look like (figure is too small) some chemical changes. Some descriptions are necessary and important to understanding the data. The reactions likely affect deformation. Also, L438: I am confused about this paragraph. How so the results imply the supply of cations are rate-limiting? Are you saying the reaction was slower when there was less Mg and Ca? It seems the carbonation tends to be limited by HCO3 (equation 6) to me? Please explain the reasoning here.*

We stated in Line 465 that we did not observe direct evidence of carbonate precipitation. These tests mainly focus on the influence of fluid - rock interaction with fluids of different compositions, we are currently working on ways that will enhance mineralization so that the effects are easier to observe in post-mortem analysis.

Comparison of the $Mg^{2+}$ and $Ca^{2+}$ concentrations measured in the $H_2O+CO_2$ open experiment and those in the $H_2O$ and $H_2O+CO_2$ close experiments show that the $Mg^{2+}$ and $Ca^{2+}$ concentrations in $H_2O+CO_2$ open experiment is low. We interpret this as a result of the consumption of $Mg^{2+}$ and $Ca^{2+}$ in the formation of carbonate. This indicates that the supply of $CO_2$ is sufficient in the $H_2O+CO_2$ open experiment and it is the supply of cations that limits the precipitation. If the low $Mg^{2+}$ and $Ca^{2+}$ concentrations are only caused by the presence of $HCO_3^-$ we would expect the $H_2O+CO_2$ close to show similar low $Mg^{2+}$ and $Ca^{2+}$ concentrations rather than being more like the $H_2O$ experiment.

- *L 366-378 There is no localization in the dry experiment (no failure) so I do not understand how these results are consistent the previous studies cited? Which specifically looked at localization associated with failure? Figure 11a shows very little localization. (L384 – If that is true, then why didn't the sample localize and fail?)*

Although the final failure is not observed in the dry experiment, we have observed temporarily stress drop with associated AE recordings. The sample did not show final failure as the experiment is halted earlier due to the failing of strain gauges and LVDTs.

Figure A8 shows the localization better which transects the sample in the middle.

If the deformation is highly localized during the creep, we would expect a clearly defined fault plane rather than the complex network of fracture shown in Figure A8.

- *L433 I think it makes more sense to say that dissolution weakens the rock, rather than the dissolution strengthens the rock (it is not stronger than dry, that we know).*

We have rephrased the statement to '…indicating that dissolution associated with fluid presence weakens the rock while precipitation reactions slightly strengthen the rock and partly compensate the effect of dissolution'. (Line 463-464)

*Minor Comments:*

*Line 187-188: Describing the phases as rapidly or slowly evolving and varying does not explain what is evolving. Strain? Strain rate?*

Thank you for pointing this out, we have clarified that here we refer to the evolution of strain.

*In many figures the fonts and images are too small and I cannot easy read or process what is being presented. 1, 3, 7, 8, 11 (also the black background in not very helpful)*

We have updated these figures to be more reader friendly.

*Figure 4: The data here is presented in a way that is hard to follow and I am not convinced it adds much to the analysis.*

The Figure 4 a) is to show the power-law relationship between the accumulated creep strain during different creep phase. And Figure 4 b) shows that the relationship observed in a) is independent of stress. Figure 4 c) and d) shows the stress dependence of creep strain during each creep phase. We find this to be an important observation that we would like to keep in the paper.

*Figure 6: I recommend plotting permeability in SI units (m^2)*

We have revised the Figure 6 following the suggestion.

*Figure 9: I cannot tell the different greens apart*

We have changed the symbol for experiment OR2_T in Figure 9 to further distinguish the two $H_2O+CO_2$ experiments.

*L 223: compressive is a stress term not a strain term*

We thank the reviewer for pointing this out and have rephrased the statement to 'compactive'. (Line 244)

*L340: C' is often as low as 50 % of failure strength, although I agree 11% is low. I would argue that your definition of failure strength is not comparable to previous studies since they are defining based on constant strain rate tests. I would be careful about how this is worded because differences in how this is being defined are not clear.*

We have revised the statement. We agree with the reviewer that the term ultimate strength often defined based on constant strain rate tests. However, this would not change our argument as the strength from constant strain rate tests would be expected to only be higher and, therefore, creep would occur at an even lower percentage of ultimate strength.

*L354: Is this stress effect on amplitude only for fluid saturated conditions (not clear)? As wing cracks can also grow longer at higher differential stress and that can lead to coalescence, both of which might reflect larger events.*

The stress effect is observed in all experiments. However, stronger effect is observed in experiments fluid saturated conditions. We clarified the text accordingly. (Line 380-381)

*L361: I have never heard bulging referred to as a dilational feature. I am pretty sure it occurs during cataclastic pore collapse. Reference?*

We have revised the statement to clarify our reasoning. (Line 388-389)

*L415: I don't understand how the relationship between the stress and total strain accommodated during phase 2 can be independent of fluid conditions if the strain rate is dependent on fluid conditions.*

Here we are implying that the ratio between the accumulated creep strain during phase I and phase II is independent of fluid conditions, which is also supported by our experimental observations. We have revised the statement to be more explicit. (Line 443-444)